# Development, Implementation and Application of Confocal Laser Endomicroscopy in Brain, Head and Neck Surgery—A Review

**DOI:** 10.3390/diagnostics12112697

**Published:** 2022-11-04

**Authors:** Irini Kakaletri, Maximilian Linxweiler, Serine Ajlouni, Patra Charalampaki

**Affiliations:** 1Medical Faculty, Rheinische Friedrich Wilhelms University of Bonn, 53127 Bonn, Germany; 2Department of Otorhinolaryngology, Head and Neck Surgery, Saarland University Medical Center, 66421 Homburg, Germany; 3Medical Faculty, University Witten Herdecke, 58455 Witten, Germany; 4Department of Biochemistry, University of Florida, Gainesville, FL 32611, USA; 5Department of Neurosurgery, Cologne Medical Center, University Witten Herdecke, 51109 Cologne, Germany

**Keywords:** confocal laser endomicroscopy, optical technologies, neurosurgery, otorhinolaryngology

## Abstract

When we talk about visualization methods in surgery, it is important to mention that the diagnosis of tumors and how we define tumor borders intraoperatively in a correct way are two main things that would not be possible to achieve without this grand variety of visualization methods we have at our disposal nowadays. In addition, histopathology also plays a very important role, and its importance cannot be neglected either. Some biopsy specimens, e.g., frozen sections, are examined by a histopathologist and lead to tumor diagnosis and the definition of its borders. Furthermore, surgical resection is a very important point when it comes to prognosis and life survival. Confocal laser endomicroscopy (CLE) is an imaging technique that provides microscopic information on the tissue in real time. CLE of disorders, such as head, neck and brain tumors, has only recently been suggested to contribute to both immediate tumor characterization and detection. It can be used as an additional tool for surgical biopsies during biopsy or surgical procedures and for inspection of resection margins during surgery. In this review, we analyze the development, implementation, advantages and disadvantages as well as the future directions of this technique in neurosurgical and otorhinolaryngological disciplines.

## 1. Introduction

Complete tumor resection is of utmost importance and a deciding factor in the treatment of numerous tumors, even if surgery is followed by adjuvant therapy. Although complete tumor resection is the aim of the surgeon, it may fail due to limited recognition of tumor cells at the resection side. Despite adjuvant therapies, incomplete resection of a tumor poses a risk for tumor recurrence, decreased quality of life and shortened lifespan [1,2]. 

Intraoperative diagnosis of tumor cells and the definition of tumor borders are based on a variety of visualization methods as well as on the histopathologic examination of a limited number of biopsy specimens in the form of frozen sections. Frozen sections are typically used to define and differentiate tumor cells ex vivo. They are removed during the resection process. Current surgical visualization tools include surgical “microscopes”, laparoscopes, ultrasonography, intraoperative MRI and CT, as well as more sophisticated navigation devices. 

However, none of these can differentiate tumor tissue from normal tissue at a cellular level. An important distinction is necessary to achieve higher accuracy in malignant surgical tumor therapy.

Moreover, intraoperative histopathology shows several shortcomings and many biopsies are inconclusive. Firstly, the tissue architecture of the tumor can be altered and mechanically destroyed during specimen preparation. Secondly, due to tissue heterogeneity, sampling errors are possible. Thirdly, the lack of real-time interactivity with the pathologist and a waiting time of about 30 min for the results disrupt the surgical workflow. In summary, optimal surgical therapy includes the combination of maximal resection of diseased tissue with minimal damage to the normal tissue [1]. This is only achievable with the ability to identify cellular structures and differentiate tumors from normal functional tissue intraoperatively. Hence, there is an urgent need for new technological equipment combined with new surgical concepts.

The aim of the imaging technique called confocal laser endomicroscopy (CLE) is to provide microscopic information on tissue in real time [3]. This can only be achieved by bringing the microscope into the patient’s body through miniaturized fiber-optic probes. Like that, the common way of taking a tissue sample from the patient can be avoided. In addition, the second step of bringing the tissue sample under the microscope in order to analyze it is not necessary anymore. This technique allows a histopathologic analysis in real time with a resolution down to 1 μm. The cell’s cytoarchitecture can be visualized clearly with a 1000-fold magnification. In the year 2004, CLE was first introduced in gastroenterology and became an enrichment for this discipline since it supported the current standard endoscopy while performing optical biopsies [3,4].

CLE allows an intraoperative assessment of tissues at the cellular level, especially in the resection zone (Figure 1). This offers a wide range of advantages, especially in the field of oncological surgery [5]. The application of CLE-assisted surgery in surgical oncology comprises numerous procedures and specialties [6,7,8,9] with the goal of increasing not only the initial diagnosis but also the therapeutic options by extending the resection borders and, more importantly, protecting the functionality of normal tissue in critical areas of the human body. Therefore, we would like to focus on the use of CLE in oncological surgery, demonstrating the revolutionary potential and perspectives that this new method offers in different surgical disciplines. This interdisciplinary approach aims to bring surgical cancer therapy to the cellular level. 

This article describes the CLE-based surgical technique, its implementation in the operating theater, its advantages, its future perspectives and planned developments, in order to offer new frontiers and create milestones in brain, head and neck surgery. 

We will describe different CLE technologies available on the market with their pros and cons, focusing on adopting and implementing CLE in the daily routine. Furthermore, we will summarize the advantage of using CLE, the currently available technology and the limitations that one has to face in both disciplines using CLE. 

## 2. Technology and Methodology

The confocal microscopes that are used in laboratories have a big similarity to the CLE devices that are available for clinical use. The difference between these microscopes is that they are extremely miniaturized. Marvin Minsky was the inventor of the confocal microscope in 1957 [10]. The target was to focus white light on one single point. The specimens were examined by the movement of the white light measurement point by passing through it to get structural information about the desired area in order to acquire both the *x*, *y* plane (surface) and the *z*-direction (depth). In contrast to conventional light microscopes, where the entire specimen is illuminated, the confocal microscope illuminates only a small point whose diameter is limited by diffraction. This procedure has to be repeated until the entire specimen is scanned point by point. A computer collects the information and reconstructs the image. The main principle of the confocal microscope is a pinhole that is positioned in the beam path of the detected light, whose mission is to block the light coming out of the focal plane. On the one hand, it leads to a reduction of the depth of field, but in return, the resolution along the optical axis (*z*-direction) gets improved. 

In the following, we will analyze the available confocal systems approved for medical diagnosis and surgical purposes. 

### 2.1. Cellvizio 400 and 800

The confocal laser endomicroscopic system Cellvizio^®^ (Mauna Kea Technologies, Paris, France) consists of a blue laser-scanning unit (LSU-488), an infrared laser-scanning unit (LSU-800), imaging mini-optical probes and the corresponding software. 

The micro, mini-optical probe of the Cellvizio^®^ system is composed of thirty thousand optical fibers. Various confocal probes (Gastroflex™, GastroFlex™ UHD, ColoFlex™, ColoFlex™ UHD, CholangioFlex™, UroFlex™, CystoFlex™, AQ-Flex™ 19, CystoFlex™ UHD all manufactured by Mauna Kea Technologies, Paris, France) are available and can be used according to the clinical need. The lateral resolution of each probe is 1 µm with a confocal image field of view of 240 µm with a maximum 55–65 µm depth. For real-time imaging, a 4 kHz oscillating mirror for horizontal line scanning and a galvanometric mirror for frame scanning are integrated. The frame rate is 12 images per second. The blue laser uses an excitation wavelength of 488 nm and a light emission detector of 500–650 nm. The red laser uses an excitation wavelength of 785 nm and a light emission detector of 800–810 nm. After the calibration of Cellvizio^®^, the endomicroscope is ready to use. A foot pedal allows to start and stop videos. With the Cellvizio^®^ software the videos can be exported and modified.

### 2.2. Five-1 and Five-2

Five-1 technology (Optiscan Pty Ltd., Notting Hill, VIC, Australia) is an incorporation of a conventional gastroscope (Pentax, Tokyo, Japan) and the neurosurgical tool Convivo (Zeiss Meditec, Oberkochen, Germany). Only a single optical fiber is utilized for the excitation source as well as for the detection pinhole. There is a flexible connection of the probe to the laser source and the detection and image processing CPU. The excitation wavelength used by the solid-state blue laser is 488 nm. In comparison to that, the detection of light emission is at 505–585 nm. The laser beam is navigated by an electromagnetically actuated control system offering resonant scanning in both *x*- and *y*-axes and is directed into the specimen via a miniature objective lens. Lateral resolution is close to 0.7 µm. *Z*-axis actuation is achieved by an electrically controlled shape-memory alloy that enables the scanning of the tissue as deep as 250 µm from the imaging window in 4 µm increments. Vertical resolution (“optical slice thickness“) is approximately 7 µm. Each raster-scanned image is a horizontal optical section of 500 × 500 µm in size. The miniaturization helps to integrate the CLE imaging head into a handheld rigid endoscope probe with an outer diameter of 7 mm. That way the probe is well suited for endoscopic diagnosis or surgical applications. The rigid probe is available on the market in different lengths and identically constructed probes have been integrated into standard gastroenterological endoscopes, which are well established in clinical practice. Furthermore, the collection of serial images is achievable at a scan rate of 0.8 frames per second with a resolution of 1024 × 1024 pixels. In addition, a rate of 1.6 frames per second at 1024 × 512 pixels is possible. This leads to an approximation of a 1000× magnification on a screen. If the examiner changes to a review mode, it is possible to magnify details in a digital way up to 10,000 times with image software while the examination is still going on and without any interruption. 

### 2.3. Other Clinically Available Confocal Systems

Another group of CLE devices is the Vivascope family 1500, 2500 and 3000, (Caliber ID, Rochester, NY, USA) available only for cellular imaging of the skin and the HRT-RCM -I, -II -III family (Heidelberg Retinal Tomograph—Rostock Corneal Module, from Heidelberg Engineering, Heidelberg, Germany) for confocal imaging of the cornea used in ophthalmology. The confocal technology used by both groups is based on a confocal reflectance mode without the special preparation of fluorescent agents. 

There are numerous fluorescent dyes used for confocal imaging. In this review, we will focus on those that have been approved for clinical use. Different dyes have approval for different indented uses. The following subsections will analyze each with regard to the specialty they are used in. 

## 3. CLE in Neurosurgery

CLE of neurosurgical disorders, such as intracranial neoplasms, has only recently been suggested to contribute to both (A) immediate tumor characterization and detection as a completion tool to surgical biopsies during stereotactic biopsy or neurosurgical procedures, and (B) inspection of resection margins during neurosurgery. 

The first studies were performed on rodents. Sankar et al. [11] first assessed the use of miniaturized handheld confocal microscopes on mice in an experimental glioblastoma model. In this study, the authors used intravenous fluorescein and topical acriflavine as contrast agents. They performed systematic biopsies on tumoral and nontumoral tissue including the margins and came to the conclusion that confocal microscopy could easily identify the different types of tissues. In addition, it may be a good tool to assist neurosurgeons when it comes to the detection of infiltrative brain tumor margins during surgery. A great advantage can also be the fact that the sampling error can be avoided during the procedure of taking biopsies of heterogeneous glial neoplasms. This huge potential could lead to the supplementation of conventional intraoperative frozen-section pathology. 

In 2014, Georges et al. [12] evaluated the use of label free CLE assessment of glioma biopsies and came to the result that the image quality was high and cellular structures were very well differentiated. Martirosyan et al. [13] evaluated the use of a variety of rapid-acting fluorophores in providing histological information on gliomas, tumor margins and normal brain in animal models to assess the boundary of the infiltrative tumor. In vivo CLE imaging was assessed with indocyanine green (ICG), fluorescein sodium (FNa), 5-aminolevulinic acid (5-ALA), acridine orange (AO), acriflavine (AF) and cresyl violet (CV). They concluded that macroscopic fluorescence was effective for gross tumor detection, but near-infrared (NIR) CLE performed using ICG enhanced the sensitivity of tumor detection, providing real-time true microscopic histological information precisely related to the side of imaging because NIR CLE performed using ICG revealed individual tumor cells and satellites within peritumoral tissue. They also concluded that CLE provided rapid histological information precisely related to the side of microscopic imaging with imaging characteristics of cells related to the unique labeling features of the fluorophores.

Foersch et al. 2012 used CLE in order to study multiple C6 glioma cell line allografts implanted into the brain of healthy Wistar rats in vivo [14]. In that way, (A) general feasibility was demonstrated by using allografts expressing endogenous fluorescence without any dyes, and (B) a variety of fluorescent agents were examined in wild-type C6-glioma allografts. Attention was given to the distinction between healthy tissue and tissue changes. For the evaluation of further clinical application and in order to develop a set of endomicroscopic criteria, they took fresh resection specimens of various types of intracranial tumors for direct confocal endomicroscopic imaging ex vivo. A histomorphologic discrimination between neoplastic and healthy tissue was reached both in vivo and ex vivo due to characteristic fluorescent staining patterns and some unique morphologic features.

In 2013, Peyre et al. [15] compared CLE in a mouse model of aggressive meningiomas with corresponding histology images. They concluded that CLE imaging could reliably provide images of meningothelial and fibroblastic mouse meningiomas as well as of malignant meningiomas. These images correlated with the findings of the pathologist. Due to this imaging method, a sharp definition of the border between the brain and the tumor could be shown, and the identification of embedded nerves and vessels was possible. Moreover, the extension of tumors along Virchow–Robin spaces into the adjacent brain was observed with CLE in all mouse models that were used in this study.

Starting in 2011, there have been several studies on human brain tumors:

In 2011, Sanai et al. [16] published the first feasibility study in human brain tumors with intraoperative CLE. They analyzed the most frequent brain tumors (gliomas, metastasis and meningiomas), as well as less frequent ones (hemangioblastomas, central neurocytomas). They concluded that intraoperative CLE was a feasible technology for the resection of human brain tumors. Preliminary analysis demonstrated reliability for a variety of lesions in identifying tumor cells and the tumor–brain interface. In addition, they used 5-ALA with CLE and reported that intraoperative CLE could visualize cellular 5-ALA-induced tumor fluorescence within low-grade gliomas (WHO I and II) and at the brain tumor, whereas conventional methods for 5-ALA tumor fluorescence detection (microscopy with filter) did not show any sign of fluorescence. 

Charalampaki et al. 2015 used CLE for the first time integrated into neurosurgical treatment [17]. The goal was to figure out the best technical considerations needed for performing CLE in neurosurgery. They also investigated how CLE can be a part of the neurosurgical daily workflow in the operating theater as an ex vivo diagnostic module and tried to integrate the CLE technique in an easy way into the neurosurgical daily routine. To achieve this aim they used endoscopic and microscopic settings and provided an immediate and intraoperative histopathologic diagnosis of the entire entity in real time. Furthermore, they explored the best conditions for an evaluation using CLE for in vivo diagnosis of different types of intracranial and intraspinal neoplasia. They used fluorescein in vivo but stopped very soon after four cases because of an insufficient distribution of fluorescein in the intracellular space. They continued with ICG, which offers much better intraoperative CLE images because of the advantages of ICG to penetrate deeper into the tissue, to highlight only the cell cytoplasm and to overcome the hemoglobin fluorescence so that erythrocytes were not visible anymore (Figure 2). 

Eschbacher et al. used CLE ex vivo in random tumors (gliomas, schwannomas, meningiomas, hemangioblastomas, etc.) [18]. Eighty-eight regions were visualized with CLE, and corresponding biopsy samples were examined with routine neuropathological analysis. The pathologist working in a blinded fashion reviewed a subset of the images in a further evaluation of the usefulness of the device as a diagnostic tool. They concluded that intraoperative confocal imaging was very well correlated with corresponding traditional histological findings, including the identification of many pathognomonic cytoarchitectural features of various brain tumors. In the blinded study, 92.9% of the tumors were diagnosed correctly. 

Most recently, Pavlov et al. demonstrated the feasibility of intraoperative in vivo pCLE both in surgery and in the biopsy of gliomas [19]. In their prospective observational study, two contrast agents were used: 5-aminolevulinic acid or intravenous fluorescein. A 0.85 mm probe was used for stereotactic procedures, modified with the biopsy needle to have a distal opening. While performing open brain surgery, a 2.36 mm probe was used. The neurons’ autofluorescence in the cortex of the brain was observed. CLE images permit a clear distinction between healthy tissue and pathological tissue in open surgery and stereotactic biopsy using fluorescein. A huge difficulty was the establishment of 5-aminolevulinic acid confocal patterns. Neither intraoperative complications related to pCLE nor the use of either contrast agent was observed. Initial feasibility and safety of intraoperative pCLE were examined during stereotactic biopsy procedures and primary brain tumor resection. It was shown that pCLE of brain tissue is applicable to surgical guidance during operations, to improve the biopsy yield and to optimize the resection of glioma that could be achieved with the analysis of tumor margins. 

Acerbi et al. studied the ex vivo fluorescein-assisted CLE technique on a blind comparison of CLE pictures to frozen sections [20]. They found a high correlation score with correct diagnosis and concluded that CLE can be a future complementary technique for intraoperative diagnosis in glioblastoma surgery. Furthermore, Restelli et al. described in their 2021 review the correlation between CLE and classic histopathological pictures for different kinds of tumors located in the brain [21]. They concluded that histopathological intraoperative diagnosis as well as tumor margins can be very well differentiated with the use of CLE techniques. 

Cheyouo et al. evaluated the cytoarchitecture of the cerebellum and the substantia nigra by testing the usefulness of unenhanced near-infrared confocal laser reflectance imaging [22]. For this study, two fresh human cadaver brains were examined by utilizing a confocal near-infrared laser probe. They demonstrated the reliability of unenhanced near-infrared reflectance imaging when it comes to the identification and distinction of neurons and the cytoarchitecture in detail of two regions in the human brain such as the cerebellum and substantia nigra by using fresh human cadaver brain.

Charalampaki et al. 2019 used CLE as an assisted surgical technology for multifluorescent microscopy. They showed that both the number of options for real-time diagnostic imaging and the therapeutic options could be augmented. This is achievable if the resection borders of cancer are extended at a cellular level and, more essentially, if the functionality of normal tissue is protected in the adjacent areas of the human brain [23].

Cui et al. 2020 presented a compact label-free and contactless reflectance confocal microscope with a 20 mm working distance that provided <1.2 µm spatial resolution over a 600 µm × 600 µm field of view in the near-infrared region [24]. Results showed great potential for the proposed system to be translated into use as a next-generation label-free and contactless neurosurgical microscope.

In addition, automatic tissue characterization with pCLE would support the surgeon not only with the determination of the diagnosis but also with guidance during robot-assisted intervention procedures.

A deep learning-based framework was proposed by Li et al. to be able to define brain tissue for context-aware diagnosis support in oncology in neurosurgery [25]. They demonstrated how this deep learning framework can be applied in a way so that glioblastoma and meningioma tumors can be classified based on endomicroscopic data. In consideration of the results, the proposed image classification framework led to a significant improvement compared to state-of-the-art feature-based methods. Furthermore, the classification performance is ameliorated by using video data with an accuracy of 99.49%. We summarized the aspects of the most important intraoperative studies performed with the CLE technique [16,17,19,20,23,26,27,28,29,30,31,32,33] and their results in Table 1.

CLE allows the neurosurgeon not only to determine the tumor biology and the tumor margins directly but also to distinguish between normal and pathological tissue. Apart from that, it allows the visualization of important accessory structures such as nerves and vessels, which in return permits a precise resection and an assured conservation of functionality.

## 4. CLE on Ent Applications

Squamous cell carcinomas of the head and neck (HNSCCs) represent one of the six most common cancer types worldwide, accounting for about 5% of all human malignancies [34]. The majority of patients are initially treated with surgery [35]. Thereby, complete tumor resection is highly important for patient outcome and is routinely controlled during the operation by rapid section histology [36]. However, this technique delays the surgical procedure and requires an additional tissue resection in order to obtain a histological verification of cancer-free resection borders. For histological diagnosis, another operative procedure combining an endoscopic examination of the upper aerodigestive tract and a biopsy of the suspicious mucosal region is needed prior to the tumor resection itself. Against this background, CLE is a promising technique not only for the visualization and differentiation between healthy mucosa, benign lesions, dysplasia and invasive cancer but also for the identification of tumor borders and the verification of R0 resections in the management of head and neck cancer. 

Though numerous studies have confirmed the potential benefit of CLE for noninvasive real-time histological imaging in gastroenterology [37,38,39,40,41,42,43], gynecology [44,45], urology [46,47], pneumology [48,49,50], and neurosurgery [11,12,13,14,15,16,17,18,19,20,21,22,23,51] over the past years, there is a limited number of studies with small patient cohorts in the field of head and neck surgery [52,53]. White et al. first described the applicability of CLE in the head and neck region in 1999 using six healthy control tissue specimens and described its morphological similarity with corresponding H&E-stained microscopic sections [54]. This first description of CLE imaging in the head and neck area was followed by numerous in vitro [55,56,57,58,59,60] and in vivo studies [61,62,63,64,65,66,67,68,69,70,71,72,73,74,75,76,77,78,79,80,81,82,83,84,85,86,87,88,89,90,91] focusing on the noninvasive detection of head and neck cancer using CLE. Table 2 gives an overview of the most important in vitro and in vivo studies on CLE applications in the head and neck region over the past 23 years. While in most of the in vitro as well as the first in vivo studies, tissue autofluorescence seemed to be adequate to provide valuable CLE images [54,55,56,58,60,61,65,67,69,70], an optimized tissue contrast was reported in other trials using topically or systemically applied fluorophores, including acriflavine, 5-aminolevulinic acid, fluorescein, proflavine or hypericin [57,59,62,63,64,66,68,70,72,73,74,76,77,78,79,80]. From 2016 on, systemically applied fluorescein (i.v. application) was used in nearly all published CLE in vivo studies in the head and neck region and presumably provides the best tissue contrast and quality of generated CLE images [81,82,83,84,88,89,90,91]. Further studies have to show if other fluorescent dyes, e.g., indocyanine green, can potentially enhance tissue contrast even more. 

Most of the studies published on CLE in the head and neck region addressed the applicability of CLE for real-time imaging of histological tumor architecture compared to healthy or dysplastic head and neck tissue and reported a sensitivity and specificity for the differentiation between neoplastic and tumor-free tissue ranging from 45.5% up to 100% and from 40% up to 100%, respectively [65,76,79]. Nearly all studies confirmed a good correlation between CLE morphology and histology and stated a relevant benefit for the intraoperative evaluation and the surgical management of suspicious mucosal lesions [54,68,72]. The most relevant morphological criteria for accurate differentiation between tumor and healthy tissue include tissue homogeneity, cell size, borders and clusters, intraepithelial capillary loops, atypical vessels, and the nucleus/cytoplasm ratio [81,82,89,90,91]. Based on these observations, Sievert et al. developed CLE imaging scores assessing different combinations of the aforementioned morphological characteristics for laryngeal, pharyngeal, and oral cavity squamous cell carcinomas [81,82,91].

In addition to correct identification of tumor tissue, few studies addressed the question if CLE can be used to confirm tumor-free resection borders in head and neck cancer surgery, which would be highly beneficial in the surgical management of head and neck cancer as an alternative to frozen-section histology. In those studies, tumor tissue as well as tumor resection borders were evaluated with a CLE probe after intravenous application of fluorescein during the surgical procedure and compared with the results of H&E histology [83,84]. Reported accuracy, sensitivity, specificity, negative predictive value, and positive predictive value for a correct assessment of resection borders ranged from 80% to 86%, 72% to 90%, 79% to 88%, 76% to 82%, and 86% to 88%, respectively. Only one study addressed the question if CLE can be used to detect the border between cancer and adjacent non-neoplastic tissue. In this study, Linxweiler et al. investigated specimens from 135 HNSCC patients and 50 healthy controls and could show that, with the help of an experienced pathologist, tumor borders can correctly be identified in 97% of cases [60] (Figure 3). One study also investigated the potential use of CLE for noninvasive real-time imaging of sinonasal lesions and reported safe applicability with excellent imaging quality [85]. 

These studies illustrate that CLE imaging represents a highly promising technique for head and neck surgery providing real-time optical biopsies not only for the evaluation of suspicious mucosal lesions but also to potentially improve the quality of surgical treatment in terms of R0 resections by better identification of tumor borders and correct assessment of resection margins. Clinical studies addressing this potential application of CLE during tumor resections including higher patient numbers must show if one can reach a relevant benefit in patients’ functional outcome and survival compared with conventional rapid section histology, ideally in a prospective, randomized, controlled study design. 

A major limitation of all studies published so far on the in vivo application of CLE in the head and neck region is the comparably low number of patients with a maximum of 44 patients. Moreover, it has to be considered that systemically applied fluorescein is not approved by the Food and Drug Administration (FDA) and the European Medicines Agency (EMA) for a clinical application separate from diagnostic angiography and angioscopy of the eye vasculature, although the risks of severe adverse events seem to be low [38,80]. Apart from that, one has to keep in mind that there is no CLE microprobe available specifically designed for application in head and neck surgery so far, with tumors frequently located at hardly accessible areas, e.g., the hypopharynx or larynx. However, the Cellvizio^®^ GastroFlex^TM^ microprobe, which was used in most head and neck studies is quite flexible, with a diameter of only 2.7 mm, and therefore is supposed to be maneuverable enough to reach any anatomical area of the upper aerodigestive tract. 

As various in vivo studies published on CLE application in the head and neck region have proven that CLE is an applicable tool in an intraoperative setting without resulting in a relevant elongation of operation time or causing any harm to the patients, future trials will have to address relevant endpoints, e.g., a higher rate of R0 resections or better protection of healthy tissue adjacent to the tumor with a better functional outcome for the patients in order to promote the use of CLE in clinical routine. Another so far sparsely addressed future application of CLE could be the real-time analysis of molecular biomarkers, as first studies have proven that antibody-linked fluorophores are able to specifically detect their targets and can be visualized with CLE in vivo [41,42,78]. Additionally, initial studies developed automated analysis tools for CLE images of oral tissue using distance map histograms [73] or deep learning technologies [65]. Further studies enrolling a higher number of patients will have to substantiate these promising results. 

## 5. Limitations of CLE

From our 12 years of experience on CLE techniques, it is important to mention that imaging with CLE and histopathology in vivo have some limitations that are mentioned below. 

(1)There are some fluorescent agents for instance cresyl violet and acriflavine that have no approval for their use in a clinical setting in neurosurgery. For this reason, we prefer fluorescein and indocyanine green for intravenous application, since the use of both fluorescent agents is well known for a lot of years now in clinical practice. Systems that are working label free will be the best outlook for the future.(2)Even if we found out during our observations that the strongest signal of the applied fluorescent agent was on the surface of the tumor, the infiltration depth of the endomicroscope is limited, which could represent a mentionable disadvantage. Nevertheless, the next generation of confocal systems (e.g., with near-infrared probes) could possibly provide a solution to this problem. Furthermore, confocal systems which have numerous excitation wavelengths, will make clinical use easier in the future.

## 6. Conclusions

The application of CLE-assisted surgery in oncologic surgery includes plenty of procedures and specialties with a lot of common goals. First of all, both the initial diagnostic accuracy has to be increased, and the therapeutic options can be augmented if the resection borders are extended. Secondly, the functionality of normal tissue in critical areas of the human body has to be protected.

In our research, we tried to demonstrate this new method that revolutionizes and influences with its great potential and perspectives not only the diagnosis but also the treatment of several pathologies that are known in various regions of the brain, head and neck. In summary, this paper gives a description of CLE as a surgical technique and demonstrates its implementation in the operating theater, its advantages, its potential in the future and its progress in different oncological surgical fields. In conclusion, it can be said that CLE-assisted surgery not only improves the representation of histological diagnosis significantly compared to other methods such as the hematoxylin and eosin staining that are time consuming and take several days but also identifies the borders between cancer and healthy tissue, which leads to maximization of cancer resection.

## Figures and Tables

**Figure 1 diagnostics-12-02697-f001:**
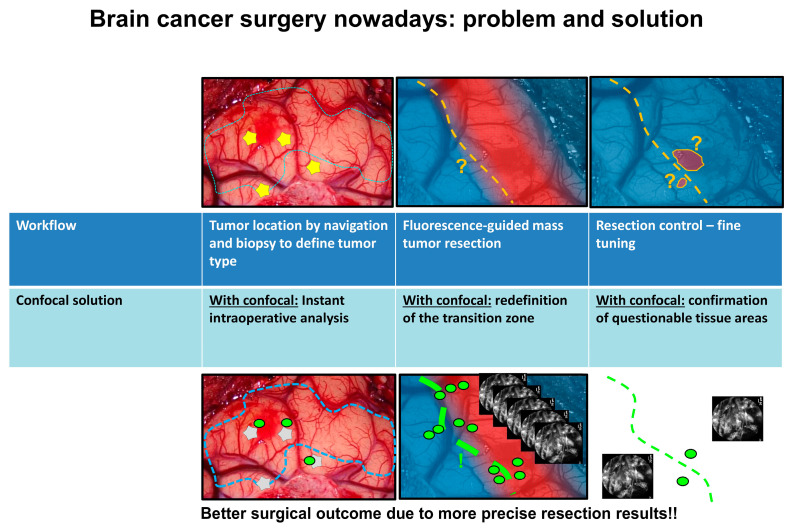
The problem faced in neurosurgical tumor treatment and how CLE could help to increase the intraoperative visualization perspectives. The green dots represent the confocal scanning. The lines show the resection borders.

**Figure 2 diagnostics-12-02697-f002:**
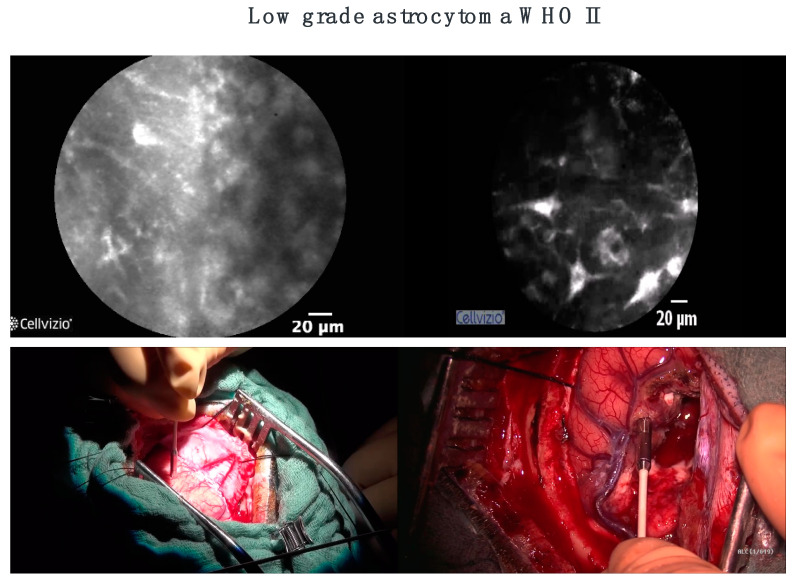
State-of-the-art visualization of the brain today is 10-fold magnification without cell visualization (pictures **below**). With CLE, we can see tumor borders (above **left**) and tumor-free zone at the end of the surgery (above **right**).

**Figure 3 diagnostics-12-02697-f003:**
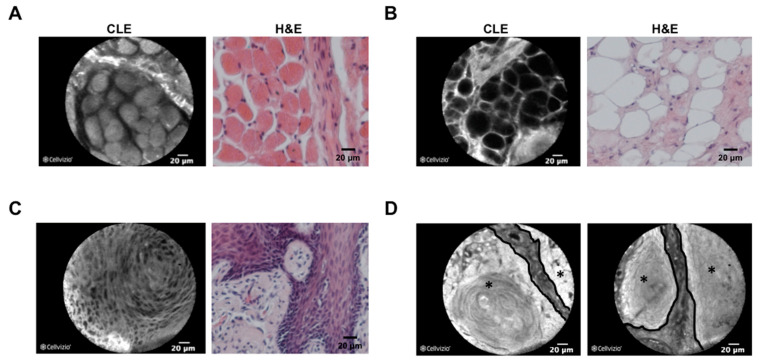
Comparison between CLE and H&E morphology of tissue in the head and neck area. Confocal laser endomicroscopy (CLE) images (left picture, fresh frozen samples) and H&E-stained slides from the respective FFPE tissue are shown for skeletal muscle of the soft palate (**A**), adipose tissue of the cheek (**B**), a nonkeratinizing squamous cell carcinoma of the tonsil (**C**) and a keratinizing squamous cell carcinoma of the tongue (**D**). As shown in (**D**), tumor borders (black line), as well as tumor localization (black star), can clearly be identified using CLE technology. CLE and H&E images are shown in 40× magnification.

**Table 1 diagnostics-12-02697-t001:** Intraoperative use of CLE in neurosurgery.

Name of the Study	* Date *	* Author *	* Number of * * Patients *	* Stain Used *	* Results *
* **Laser-Scanning Confocal Endomicroscopy (LSCE) in the Neurosurgical Operating Room: a review and Discussion of future applications** *	2011	Sanai, N. [16]	10	Fluorescein 5-ALA	- There are technical limitations in visualization. Specifically, in regard to the identification of nuclear morphology, cytoplasm characteristics and nuclear-to-cell ratios
* **Comparing High-Resolution Microscopy Techniques for Potential Intraoperative use in Guided Low-Grade Glioma Resections** *	2015	Meza, D. [26]	7	5-ALA	- Out of the three techniques investigated, only DAC microscopy was able to offer high image resolution comparable to histology - Disadvantage of DAC: requires a laser-scanning mechanism to achieve sectioning
* **Confocal Laser Endomicroscopy for Real-time Histomorphological Diagnosis: Our Clinical Experience with 150 Brain and Spinal Tumor Cases** *	2015	Charalampaki, P. [17]	150	Acriflavine, Fluorescein	- Key features of different types of brain and spinal tumors were observed - CLE offered benefits for patients, neurosurgeons and other oncological disciplines - In general, waiting times for histopathological review have been greatly reduced using CLE
* **Intraoperative Probe-Based Confocal Laser Endomicroscopy in Surgery and Stereotactic Biopsy of Low-Grade and High-Grade Gliomas: A Feasibility Study in Humans** *	2016	Pavlov, V. [19]	9	5- ALA (3) FNa (6)	- Cellvizio allowed for successful differentiation between healthy and pathological tissue - The intraoperative dyes do not pose any intraoperative risks - Fluorescence allows better differentiation between tissue types
* **Prospective Evaluation of the Utility of Intraoperative Confocal Laser Endomicroscopy in Patients with Brain Neoplasms using Fluorescein Sodium: Experience with 74 Cases** *	2016	Martirosyan, NL. [27]	74	FNa	- Mean duration of intraoperative CLE was approximately 16 min - Glioma specificity and sensitivity were 94% and 91% - Meningioma specificity and sensitivity were 93% and 97% - CLE could allow for interactive identification of tumor areas and improve tumor resection
* **Visualization of Brain Microvasculature and Blood flow** * **in Vivo** * **: Feasibility Study using Confocal Laser Endomicroscopy** *	2021	Belykh, E. [28]	20	FNa	- CLE allows for precise details of real-time cell movements. - CLE allowed for classification of the microvasculature in gliomas into normal and abnormal. - Microvasculature was visible for up to 30 min after 2 mg application.
* **Probe-Based Three-Dimensional Confocal Laser Endomicroscopy of Brain Tumors: Technical Note** *	2018	Belykh, E. [29]	Mice: 19 Patients: 31	FNa	- 3D-rendered images allow an increased spatial understanding of cellular architecture and structures
* **Confocal-Assisted Multispectral Fluorescent Microscopy for Brain Tumor Surgery** *	2019	Charalampaki, P. [23]	Rats: 22 Patients: 13	ICG	- Optimizing microsurgery by improving its safety and efficiency is an important aspect of oncological neurosurgery - There are different techniques that are being investigated in order to determine the best approach - A technique of importance is the use of ICG and CLE to help contrast tumors and tumor margins - There are still limitations to overcome; however, CLE proves promising
* **Intraoperative Confocal Laser Endomicroscopy** * **Ex Vivo** * **Examination of Tissue Microstructure During Fluorescence-Guided Brain Tumor Surgery** *	2020	Belykh, E. [30]	47	FNa	- Specificity of CLE: 90% - Diagnostic features were able to be identified and allowed for a distinction between healthy brain tissue and lesional gliomas
**Ex Vivo** * **Fluorescein-Assisted Confocal Laser Endomicroscopy (CONVIVO System) in Patients with Glioblastoma: Results from a Prospective Study** *	2020	Acerbi, F. [20]	15	FNa	- Blind comparison between CONVIVO-Zeiss and frozen image showed a high correlation of correct diagnosis. - CLE can be used as a complementary tool for intraoperative diagnosis during glioblastoma surgery
* **Intraoperative Imaging of Brain Tumors with Fluorescein: Confocal Laser Endomicroscopy in Neurosurgery. Clinical and User experience** *	2021	Höhne, J. [31]	12	FNa	- CLE allowed high-quality visualization of fine structures, anatomical details and microstructures. - Comparative validation from neuropathologists is crucial for CLE success
* **Intraoperative Confocal Laser Endomicroscopy: Prospective** * **in Vivo** * **Feasibility Study of a Clinical-Grade System for Brain Tumors** *	2022	Abramov, I. [32]	30	FNa	- Interpretable CLE results were attained in 7 min - Positive correlation between interpretable image acquisition and surgeon experience, cumulative length of CLE time and CLE time per case - Accuracy, sensitivity and specificity were at 94%, 94%, and 100%, respectively - The safety and feasibility of obtaining noninvasive biopsy results were proven successful
* **Real-Time Intraoperative Surgical Telepathology using Confocal Laser Endomicroscopy** *	2022	Abramov, I. [33]	11	FNa	- CLE identification has proven to be faster than neuropathological analysis (6 min vs. 23 min) - CLE does not only allow quicker analysis but also allows a less invasive approach to pathology by allowing pathologists to view the images simultaneously with the surgeons

CLE: confocal laser endomicroscopy; FNa: sodium fluorescein; 5-ALA: aminolevulinic acid; GFP: green fluorescent protein; MG: magnifying glasses; DAC: dual-axis confocal microscopy.

**Table 2 diagnostics-12-02697-t002:** CLE studies head and neck region.

	Study	Year	No. of Cases/Samples	Fluorescent Dye	Main Results
**ex vivo**	Clark et al. [55]	2003	17	none	HNSCC patients; good visualization of tumor morphology as well as adjacent tumor-free tissue
	Just et al. [56]	2006	26	none	larynx biopsies (healthy, dysplasia, benign + malignant tumors); good correlation with histology; primary endpoints Se/Sp
	Abbaci et al. [57]	2009	27	AF/F/5-ALA	laryngectomy specimens; tumor, dysplastic and healthy tissue portions of each specimen were examined; description of CLE morphology compared with HE staining
	Muldoon et al. [58]	2012	13	none	HNSCC samples; primary endpoints Se/Sp; good correlation with histology
	Vila et al. [59]	2012	38	P	HNSCC samples; 7 examiners for the evaluation of CLE images after initial training; primary endpoints Se/Sp/IRR/Ac
	Linxweiler et al. [60]	2016	185	AF/none	HNSCC samples (*n* = 135) + healthy controls (*n* = 50); visualization and discrimination between neoplastic and non-neoplastic tissue; identification of the tumor border; evaluation of CLE images by ENT surgeons, pathologists and laymen after initial training; primary endpoint: correct identification of tumor border and tumor localization
**in vivo (M)**	Farahati et al. [61]	2010	60	none	10 healthy mice, 50 mice with chemically induced tongue cancer; description of CLE morphology; primary endpoints Se/Sp/IRR
**in vivo (H)**	White et al. [54]	1999	6	none	healthy controls; description of CLE morphology; good correlation with histology
	Zheng et al. [62]	2004	5	5-ALA	2 healthy controls; 3 tongue cancer patients; description of morphology, good correlation with histology
	Thong et al. [63]	2007	not indicated	5-ALA/F/H	tissue samples + in vivo measurements in humans and mice; good correlation with histology; differentiation between healthy tissue and tumor tissue
	Thong et al. [64]	2007	2	5-ALA/F	healthy control patient + tongue cancer patient; description of morphology; good correlation with histology
	Maitland et al. [65]	2008	8	none	HNSCC patients; description of CLE morphology; good correlation with histology
	Haxel et al. [66]	2010	5	AF/F	healthy controls; description of CLE morphology; good correlation with histology
	Pogorzelski et al. [67]	2012	15	none	HNSCC patients; development of a diagnostic score; good differentiation between healthy tissue and tumor tissue
	Thong et al. [68]	2012	6	F/H	healthy controls; description of CLE morphology; good correlation with morphology; application of a 3D fluorescence imaging prototype
	Pierce et al. [69]	2012	30	none	moderate to severe dysplasia + HNSCC patients; good correlation with histology; primary endpoints Se/Sp/PPV/NPV
	Just & Pau [70]	2013	10	none	visualization of laryngeal mucosa from healthy controls and patients with premalignant lesions
	Contaldo et al. [71]	2013	6	AF	healthy controls; visualization of different histological structures
	Nathan et al. [72]	2014	21	F i.v.	visualization of premalignant and malignant lesions of the head and neck mucosa (12 dysplasias, 9)
					(9 carcinomas); good correlation with histology; primary endpoints Se/Sp/PPV/NPV
	Dittberner et al. [73]	2016	12	F i.v.	automated analysis of CLE images from neoplastic and non-neoplastic oral tissue; primary endpoint AUC
	Moore et al. [74]	2016	24	F i.v.	visualization and discrimination between benign, precancerous and malignant lesions of the head and neck; primary endpoint interobserver agreement; good correlation with histology
	Volgger et al. [75]	2016	19	F i.v.	visualization and discrimination between healthy tissue and various grades of dysplasia up to squamous cell carcinomas of the laryngeal mucosa; primary endpoints Se/Sp; CLE helpful for the discrimination between noninvasive laryngeal lesions
	Goncalves et al. [76]	2017	7	F i.v.	visualization and differentiation between severe dysplasia to invasive carcinoma (*n* = 3) and benign tumors (*n* = 4) of the vocal cords; primary endpoints Se/Sp/PPV/NPV/IRR
	Aubreville et al. [77]	2017	12	F i.v.	automated analysis of CLE images of the cancerous and tumor-free oral mucosa from 12 HNSCC patients using a deep learning approach; primary endpoints Se/Sp/AUC
	Englhard et al. [78]	2017	11	FITC-labeled Ab	visualization and differentiation between HNSCC and tumor-free tissue using CLE in combination with FITC-labeled EpCAM and EGF-R-antibodies; in vitro (cell lines) + in vivo (HNSCC samples, *n* = 11; healthy mucosa samples, *n* = 5); primary endpoint antigen specificity of the Abs
	Goncalves et al. [79]	2019	7	F i.v.	visualization and differentiation between squamous cell carcinomas (*n* = 3) and benign tumors (*n* = 4) of the vocal folds; primary endpoints Se/Sp/PPV/NPV/IRR
	Shinohara et al. [86]	2020	10	AF Food Red No. 106	visualization of and differentiation between HNSCC and adjacent healthy tissue using autofluorescence, topical AF, or AF + Food Red No. 106; best results with AF only
	Sievert et al. [84]	2021	5	F i.v.	visualization and differentiation between oropharyngeal squamous cell carcinomas and adjacent healthy tissue; assessment of free resection margins; primary endpoints Se/Sp/PPV/NPV/Ac
	Wenda et al. [85]	2021	2	F i.v.	Visualization of tumor tissue in one patient with sinonasal inverted papilloma and one patient with sinonasal squamous cell carcinoma
	Dittberner et al. [87]	2021	13	F i.v.	visualization of and differentiation between HNSCC and adjacent healthy tissue; primary endpoints Se/Sp/Ac/; concordance between CLE imaging and histology
	Sievert et al. [82]	2021	13	F i.v.	generation and evaluation of an eight-point score for correct assessment of malignancy in laryngeal and pharyngeal squamous cell carcinoma; primary endpoints Se/Sp/Ac/NPV/PPV/AUC
	Sievert et al. [83]	2021	5	F i.v.	CLE-based assessment of safe surgical margins in laryngeal cancer patients; primary endpoints Se/Sp/NPV/PPV/Ac
	Sievert et al. [81]	2022	13	F i.v.	generation and evaluation of a larynx and pharynx confocal imaging score for correct assessment of malignancy in laryngeal and pharyngeal squamous cell carcinomas; comparison between CLE experts and CLE nonexperts; primary endpoints Se/Sp/Ac
	Abbaci et al. [88]	2022	44	patent blue V	visualization of and differentiation between HNSCC tumor core and its margins; primary endpoints Se/Sp
	Sievert et al. [89]	2022	5	F i.v.	visualization of and differentiation between tumor and adjacent healthy tissue in 5 laryngectomy patients; primary endpoints Se/Sp, ROI of tumor and healthy tissue
	Sievert et al. [90]	2022	10	F i.v.	visualization and evaluation of diagnostic value of intraepithelial capillary loops and atypical vessels in 10 laryngectomy patients; comparison between tumor vs. healthy tissue; primary endpoints Se/Sp/NPV/PPV/Ac
	Sievert et al. [91]	2022	12	F i.v.	generation and evaluation of a confocal imaging score for correct assessment of malignancy in oral cavity squamous cell carcinomas; primary endpoints Se/Sp/Ac/NPV/PPV/AUC

H—Hypericin; AF—Acriflavine; F—Fluorescein; 5-ALA—5 aminolevulinic acid; P—Proflavine; Ab—Antibodies; FITC—Fluorescein isothiocyanate; Se—Sensitivity; Sp—Specificity; PPV—Positive predictive value; NPV—Negative predictive value; Ac—Accuracy; IRR—Inter-rater reliability; AUC—Area under the curve.

## Data Availability

Data can be provided by the corresponding author upon reasonable request.

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
