# Peer review of "Development, Implementation and Application of Confocal Laser Endomicroscopy in Brain, Head and Neck Surgery—A Review"

_diagnostics, 2022, doi:10.3390/diagnostics12112697_

Round 1

Reviewer 1 Report

There are many unclear or even incorrect statements in the introduction. E.G." Frozen sections are (...)removed when the tumor is exposed but not yet resected". This is not the case in head and neck surgery. There are no citations in the introduction, although many partially questionable statements are made. 

The paper needs linguistic revision, there is a lot of repetition and redundancy. 

Main criticism is the incomplete research. Regarding the application of CLE in the ENT field, several important works have not been mentioned. Enclosed are some of them: 

Wenda N, Kiesslich R, Gosepath J. Endonasale konfokale Laserendomikroskopie – erste Anwendung und Validierung von Malignitätskriterien [Confocal laser endomicroscopy - first application and validation of malignancy criteria]. Laryngorhinootologie. 2021 Oct;100(10):818-823.   Sievert M, Stelzle F, Aubreville M, Mueller SK, Eckstein M, Oetter N, Maier A, Mantsopoulos K, Iro H, Goncalves M. Intraoperative free margins assessment of oropharyngeal squamous cell carcinoma with confocal laser endomicroscopy: a pilot study. Eur Arch Otorhinolaryngol. 2021 Nov;278(11):4433-4439.   Sievert M, Oetter N, Aubreville M, Stelzle F, Maier A, Eckstein M, Mantsopoulos K, Gostian AO, Mueller SK, Koch M, Agaimy A, Iro H, Goncalves M. Feasibility of intraoperative assessment of safe surgical margins during laryngectomy with confocal laser endomicroscopy: A pilot study. Auris Nasus Larynx. 2021 Aug;48(4):764-769. doi: 10.1016/j.anl.2021.01.005. Epub 2021 Jan 16. PMID: 33468350.   Sievert M, Oetter N, Mantsopoulos K, Gostian AO, Mueller SK, Koch M, Balk M, Thimsen V, Stelzle F, Eckstein M, Iro H, Goncalves M. Systematic classification of confocal laser endomicroscopy for the diagnosis of oral cavity carcinoma. Oral Oncol. 2022 Sep;132:105978. doi: 10.1016/j.oraloncology.2022.105978. Epub 2022 Jun 21. PMID: 35749803.   Sievert M, Mantsopoulos K, Mueller SK, Eckstein M, Rupp R, Aubreville M, Stelzle F, Oetter N, Maier A, Iro H, Goncalves M. Systematic interpretation of confocal laser endomicroscopy: larynx and pharynx confocal imaging score. Acta Otorhinolaryngol Ital. 2022 Feb;42(1):26-33. doi: 10.14639/0392-100X-N1643. Epub 2022 Feb 7. PMID: 35129541; PMCID: PMC9058938.

Author Response

Reviewer 1:

Point 1:There are many unclear or even incorrect statements in the introduction. E.G." Frozen sections are (...)removed when the tumor is exposed but not yet resected". This is not the case in head and neck surgery. There are no citations in the introduction, although many partially questionable statements are made. 

Response 1: We added citations to the introduction and revised the text.

Point 2: The paper needs linguistic revision, there is a lot of repetition and redundancy.

Response 2: We read over the whole manuscript and focused on grammar, spelling mistakes and repetitive sentences. We reduced redundancy. 

Point 3: Main criticism is the incomplete research. Regarding the application of CLE in the ENT field, several important works have not been mentioned. Enclosed are some of them: 

Wenda N, Kiesslich R, Gosepath J. Endonasale konfokale Laserendomikroskopie – erste Anwendung und Validierung von Malignitätskriterien [Confocal laser endomicroscopy - first application and validation of malignancy criteria]. Laryngorhinootologie. 2021 Oct;100(10):818-823.   Sievert M, Stelzle F, Aubreville M, Mueller SK, Eckstein M, Oetter N, Maier A, Mantsopoulos K, Iro H, Goncalves M. Intraoperative free margins assessment of oropharyngeal squamous cell carcinoma with confocal laser endomicroscopy: a pilot study. Eur Arch Otorhinolaryngol. 2021 Nov;278(11):4433-4439.   Sievert M, Oetter N, Aubreville M, Stelzle F, Maier A, Eckstein M, Mantsopoulos K, Gostian AO, Mueller SK, Koch M, Agaimy A, Iro H, Goncalves M. Feasibility of intraoperative assessment of safe surgical margins during laryngectomy with confocal laser endomicroscopy: A pilot study. Auris Nasus Larynx. 2021 Aug;48(4):764-769. doi: 10.1016/j.anl.2021.01.005. Epub 2021 Jan 16. PMID: 33468350.   Sievert M, Oetter N, Mantsopoulos K, Gostian AO, Mueller SK, Koch M, Balk M, Thimsen V, Stelzle F, Eckstein M, Iro H, Goncalves M. Systematic classification of confocal laser endomicroscopy for the diagnosis of oral cavity carcinoma. Oral Oncol. 2022 Sep;132:105978. doi: 10.1016/j.oraloncology.2022.105978. Epub 2022 Jun 21. PMID: 35749803.   Sievert M, Mantsopoulos K, Mueller SK, Eckstein M, Rupp R, Aubreville M, Stelzle F, Oetter N, Maier A, Iro H, Goncalves M. Systematic interpretation of confocal laser endomicroscopy: larynx and pharynx confocal imaging score. Acta Otorhinolaryngol Ital. 2022 Feb;42(1):26-33. doi: 10.14639/0392-100X-N1643. Epub 2022 Feb 7. PMID: 35129541; PMCID: PMC9058938.

Response 3: We added the mentioned references in the section of ENT and also to table 2 and discussed them on page 14-15 in the text. All the references can be found at the end in the reference section with the numbers 73-83.

Reviewer 2 Report

In this interesting review article, the Authors reviewed the confocal laser imaging development and use in neurosurgery and in ENT surgery. This technology has undergone an interesting process of development and diffusion among neurosurgeons in recent years, thanks to its capability to provide "real-time" intraoperative tissue diagnosis, in a similar manner to frozen section techniques.

In this particular review, differently from other similar works (see below), also the development of these machines and their progressive diffusion in neurosurgery is studied and reviewed. Also the application of this technology in ENT surgery represents an innovative adjunct, although this is out of my area of expertise. Hence, the main theme of this work is surely actual and innovative.

 The article would increase its scientific strength by addressing the following points:

 1.       I suggest adding and discussing the experience of our group with CLE at the Istituto Neurologico Carlo Besta group in Milano, Italy; the paper by Acerbi F. (2020) is just cited in table 1, but not discussed nor listed in the references. Similarly, the review by Restelli F. (Journal of Clinical Medicine, 2021) should be cited and discussed.

 2.      I would shorten the conclusions section, that is too extensive, moving some parts to the discussion section.

 3.       English written language would benefit from a mother tongue revision.

Author Response

Reviewer 2:

Point 1: I suggest adding and discussing the experience of our group with CLE at the Istituto Neurologico Carlo Besta group in Milano, Italy; the paper by Acerbi F. (2020) is just cited in table 1, but not discussed nor listed in the references. Similarly, the review by Restelli F. (Journal of Clinical Medicine, 2021) should be cited and discussed.

Response 1: We added the paper from Acerbi F. (2020) and the review by Restelli F.(Journal of Clinical Medicine,2021) and discussed them in the text on page 6-7. They are cited in the references with the number 20 and 21.

Point 2: I would shorten the conclusions section, that is too extensive, moving some parts to the discussion section.

Response 2: We shortened the conclusion section and added a new chapter called “5.limitations of CLE” on page 15.

Point 3: English written language would benefit from a mother tongue revision.

Response 3: We revised the manuscript and focused on improving the written language.

Reviewer 3 Report

1.     Please revise Figure 1, some words and sentence are too small and unclear to read. Please revise it.

2.     Please revise figure 2, the radiological image is unclear. 

3.     Why the sentence “The aim of the imaging technique called Confocal Laser Endomicrosocpy (CLE)” is marked in bold? (Page 2).

4.     Please add the reference to the sentence “In the year 2004, CLE was first introduced in gastroenterology and became an enrichment for this discipline since it supported the current standard endoscopy while performing optical biopsies”

5.     Again, please add reference to “Marvin Minsky was the first who developed the principle of the confocal microscope in the 1950s”.

6.     From 1. introduction to 2.3 other clinically available confocal systems, there’s no reference inside. Please revise it and add adequate references to certain paragraphs. 

7.     In the bottom of page 6, “Table 1” is isolated. Please revise it. 

8.     Again, in the bottom of page 10, “Table 2” is isolated. Please revise it. 

9.     Why the sentence “bold print - i.v. application of fluorescent dye” is marked in bold. Besides, what’s the title format of this sentence? 

10.  There is too much content inside the conclusion section (Page 14). please make the section concise and precise.  

Author Response

Reviewer 3

Point 1: Please revise Figure 1, some words and sentence are too small and unclear to read. Please revise it.

Response 1: We revised figure 1 on page 2 and shortened the sentences in order to make it more understandable.

Point 2: Please revise figure 2, the radiological image is unclear. 

Response 2: We revised figure 2 on page 6.

Point 3: Why the sentence “The aim of the imaging technique called Confocal Laser Endomicrosocpy (CLE)” is marked in bold? (Page 2).

Response 3: We changed the sentence “the aim of the imaging technique called Confocal Laser Endomicroscopy (CLE)” so that it is not marked in bold anymore.

Point 4: Please add the reference to the sentence “In the year 2004, CLE was first introduced in gastroenterology and became an enrichment for this discipline since it supported the current standard endoscopy while performing optical biopsies”

Response 4: We added the reference to the sentence. It is in the introduction on page 2.

Point 5: Again, please add reference to “Marvin Minsky was the first who developed the principle of the confocal microscope in the 1950s”.

Response 5: We added the reference to the sentence “Marvin Minsky was the first who developed the principle of the confocal microscope in the 1950s” on page 3 and changed the sentence to “Marvin Minsky was the inventor of the confocal microscope in 1957”.

Point 6: From 1.introduction to 2.3 other clinically available confocal systems, there’s no reference inside. Please revise it and add adequate references to certain paragraphs.

Response 6: We revised all the chapters from 1.introduction to 2.3 other clinically available confocal systems and added the references.

Point 7: In the bottom of page 6, “Table 1” is isolated. Please revise it.

Response 7: We revised Table 1 on page 7 adding the title intraoperative use of CLE in neurosurgery.

Point 8: Again, in the bottom of page 10, “Table 2” is isolated. Please revise it.

Response 8: We revised table 2 and added some more references.

Point 9: Why the sentence “bold print - i.v. application of fluorescent dye” is marked in bold. Besides, what’s the title format of this sentence? 

Response 9: We deleted the sentence and replaced in Table 2 “fluorescein i.v.” with “F i.v.”

Point 10: There is too much content inside the conclusion section (Page 14).please make the section concise and precise.

Response 10: We shortened the conclusion section and added a new chapter called “5.limitations of CLE” on page 15.

Round 2

Reviewer 1 Report

thank you at the authors. The manuscript has improved significantly and all questions were answered accordingly. I approve the acceptance of this manuscript. 

Reviewer 3 Report

The authors address most of my concerns. 

I suggest "Accept in present form".